# Interactions of Gram-Positive Bacterial Membrane Vesicles and Hosts: Updates and Future Directions

**DOI:** 10.3390/ijms25052904

**Published:** 2024-03-01

**Authors:** Giuseppe Sangiorgio, Emanuele Nicitra, Dalida Bivona, Carmelo Bonomo, Paolo Bonacci, Maria Santagati, Nicolò Musso, Dafne Bongiorno, Stefania Stefani

**Affiliations:** Department of Biomedical and Biotechnological Sciences (BIOMETEC), University of Catania, 95125 Catania, Italy; emanuele.nicitra@phd.unict.it (E.N.); dalida.bivona@phd.unict.it (D.B.); carmelo.bonomo@phd.unict.it (C.B.); paolo.bonacci@phd.unict.it (P.B.); m.santagati@unict.it (M.S.); nmusso@unict.it (N.M.); stefania.stefani@unict.it (S.S.)

**Keywords:** Gram-positive bacteria, membrane vesicles, host–pathogen interactions

## Abstract

Extracellular vesicles (EVs) are lipid bilayers derived from cell membranes, released by both eukaryotic cells and bacteria into the extracellular environment. During production, EVs carry proteins, nucleic acids, and various compounds, which are then released. While Gram-positive bacteria were traditionally thought incapable of producing EVs due to their thick peptidoglycan cell walls, recent studies on membrane vesicles (MVs) in Gram-positive bacteria have revealed their significant role in bacterial physiology and disease progression. This review explores the current understanding of MVs in Gram-positive bacteria, including the characterization of their content and functions, as well as their interactions with host and bacterial cells. It offers a fresh perspective to enhance our comprehension of Gram-positive bacterial EVs.

## 1. Introduction

The fundamental process of secreting cellular components across the plasma membrane is a universal occurrence in all life forms, facilitating interactions between organisms and their surroundings. This mechanism is achieved through the release of vesicles—spherical, nanosized structures derived from the lipid membranes of the cell surface [1,2,3]. Recently, there has been increased attention on extracellular vesicles (EVs) [4,5,6]. Originally discovered in eukaryotes, EVs from the plasma membrane carry proteins, nucleic acids, and lipids, categorized mainly into exosomes, microvesicles, and apoptotic bodies by size [7,8]. Subsequently, EVs were identified in prokaryotes, including bacteria [1,9]. Bacteria are classified into Gram-negative or Gram-positive based on membrane structure, with Gram-negative bacteria having two membrane layers separated by the periplasm. In contrast, Gram-positive bacteria possess a distinct membrane structure consisting of one membrane and a thicker layer of peptidoglycan. In Gram-negative bacteria, EVs are termed outer membrane vesicles (OMVs), while in Gram-positive bacteria, they are referred to as membrane vesicles (MVs) [10,11]. Both OMVs and MVs constitute subclasses of microbial EVs. Extensive research has been conducted on OMVs to elucidate their functions [12,13,14]. Questions about the relatively low production of vesicles in Gram-positive bacteria were prevalent in the past. However, over the last decade, substantial research has been dedicated to understanding the role of Gram-positive MVs [15]. Observations have indicated the presence of MVs in both pathogenic and nonpathogenic Gram-positive bacteria across diverse growth conditions and environments, suggesting the universal and widespread nature of MV secretion [16,17,18,19]. Depending on the packaged cargo, EVs have been implicated in pathogenesis, antibiotic resistance, stress response, intercellular competition, and nucleic acid transfer. The increase in studies on Gram-positive bacteria, such as *Staphylococcus aureus*, *Bacillus anthracis*, and *Streptococcus mutans*, has revealed that MVs in Gram-positive bacteria exhibit a 20- to 400-nm bilayer spherical structure [20,21,22]. This review explores the current understanding of the characterization of the content and functions of MVs in Gram-positive bacteria, as well as their interactions with host and bacterial cells.

## 2. Vesiculogenesis

The formation of membrane vesicles is well-documented in archaeal, Gram-negative, and mammalian cells, yet understanding MV biogenesis in Gram-positive bacteria, with their peptidoglycan-rich structure, has been challenging [1,23]. Various models for OMV biogenesis have been proposed, with genetic and biochemical analyses shedding light on the process [24,25]. Despite the relatively recent exploration of MVs in Gram-positive bacteria, with fewer studies compared to their Gram-negative counterparts, only a limited number of investigations have identified genetic factors responsible for vesicle formation [23,26]. Two hypotheses have been proposed to elucidate the mechanism of MVs crossing the barrier of the peptidoglycan. One suggestion has posited that MVs may be propelled through pores in the cell wall by turgor pressure, driven by budding from the cell membrane. Alternatively, it has been proposed that the peptidoglycan may undergo localized degradation, either due to enzymes associated with EVs or released alongside them [1]. MV release appears to initiate with the budding of the cytoplasmic membrane, with a hypotonic environment being pivotal for vesiculogenesis [27] (Figure 1). Lipidomic studies have revealed similarities between MVs and cytoplasmic membranes. However, variations in fatty acids and phospholipid content suggest that vesicle budding may occur in specific lipid-enriched membrane domains [22,28]. Recently, nanopods or nanotubes, filamentous structures facilitating cell-to-cell transfer and associated with EVs, have gained attention in bacterial research [29]. These structures, resembling eukaryotic ‘tunneling nanotubes’, were initially observed in hyperthermophilic bacteria and later identified in various bacteria, including Firmicutes, Myxobacteria, and Proteobacteria [30,31,32,33]. Nanotubes can bridge neighboring cells, promoting communication and facilitating molecular exchange [30]. Notably, some nanotubes contain a calcineurin-like protein, YmdB, essential for their formation and intercellular molecular exchange in *B. subtilis* [33].

In *Staphylococcus aureus*, a link between MV formation and the production of Phenol-Soluble Modulins (PSMs) has been established, enhancing membrane fluidity [27,34]. Following their production, MVs actively contribute to the weakening of peptidoglycan cross-links, with *S. aureus* utilizing autolysin to create pores for MV release [23]. Several Gram-positive species, including Group A *Streptococci* and *Bacillus subtilis*, exhibit the up-regulation of genes like *spo0A* and *spf*, and the transcription factor σ^B^, contributing to MV production [9,24]. Rath et al. demonstrated that *Mycobacterium tuberculosis* promotes MV formation by acting on *virR* gene expression, strongly stimulating the immune system [35]. Environmental factors, including surfactin and serum albumin, disrupt Gram-positive bacterial MVs, indicating host–bacteria interactions [9,36]. The release of MVs involves the facilitation of cell wall pores after their packaging at the cytoplasmic level. Holes in the peptidoglycan layer may be associated with prophage-encoded endolysin, compromising cell structural integrity and causing cytoplasmic content to be released as MVs [37]. Prophages, transmitted from the parental strain to its daughter cells under favorable environmental conditions, can induce the expression of lytic genes during stressful situations. This promotes the assembly of new phages that lyse the bacterial host cell, concurrently leading to the dumping of EVs into the extracellular space [19]. Proteomic analyses of MVs have detected penicillin-binding proteins (PBPs) and autolysins, suggesting that cell wall modification plays a pivotal role in vesicle release. Antibiotic treatment, specifically with β-lactam antibiotics, reduces peptidoglycan cross-linking, thereby increasing MV release [38]. While Gram-positive bacteria may share mechanisms with their Gram-negative counterparts, involving explosive cell lysis and prophage-derived endolysins under stress conditions [19,37], the specifics of MV release at division septa, as observed in Gram-negative bacteria, remain unproven for Gram-positive bacteria [39]. Membrane fluidity and peptidoglycan cross-linking have emerged as critical determinants of vesicle release in Gram-positive bacteria, providing insights into the dynamic processes underlying MV biogenesis in these organisms.

## 3. Composition of Membrane Vesicles

Despite decades of observation of OMVs in Gram-negative organisms, the precise mechanism behind their production remains incompletely understood. The enrichment or depletion of EV content compared to the cell suggests a regulated process. Gram-positive EVs display a diverse composition encompassing fatty acids, phospholipids, cytoplasmic and membrane-associated proteins, virulence factors, lipid acids, peptidoglycans, DNA, and sRNA. Omics technologies facilitate assigning specific roles to vesicles based on their content [39,40]. This sorting mechanism reflects differences in EV protein and nucleic acid compositions [41]. The presence of a protein core in EVs suggests a conserved sorting mechanism guiding protein packaging [42]. The sorting of proteins, lipids, and nucleic acids is lipoprotein-dependent, using molecular charge to determine EV composition. Evidence suggests molecules intercept the curvature caused by EV blubbing, guiding their direction and insertion into EVs.

### 3.1. Protein Cargo

Proteomic analysis has demonstrated variation in protein content among bacteria within the same genus [43,44]. The complex protein arrangement within vesicles includes membrane-associated and lumen-protected proteins [45]. Clinical strains of *S. aureus* exhibit diverse cytotoxic profiles, yet a conserved core composition includes surface proteins, transporter proteins, and PBPs [20]. Comparable findings occur in other Gram-positive bacteria like *M. tuberculosis*, *Streptococcus pneumoniae*, and *Bacillus anthracis* [9,22,46,47]. MVs have a bilayer structure with a central lumen, leading to intricate protein organization [48]. Moreover, protein distribution in MVs reflects their origin: the majority of membrane-associated proteins in MVs come from the cytoplasmic membrane, while those in the lumen are cytoplasmic proteins packaged during vesiculogenesis [41]. Some studies have focused on the fact that the MV proteome can include not only proteins from the MV lumen and membrane but also proteins associated with the MV surface [18]. Functional analysis has revealed a prevalence of metabolism-associated proteins in MVs [22,49]. The role of these non-virulence-associated proteins in MVs remains unclear [45,46]. Proteomic analyses of Gram-positive bacterial MVs highlight lipoprotein enrichment [9,49]. Lipoproteins, being Toll-like receptor 2 (TLR2) ligands, play a pivotal role in host immune responses to bacterial infection [50,51]. The presence of lipoproteins significantly affects the host immune response to MVs, influencing various aspects of bacterial growth, immune activation, and virulence [52,53].

### 3.2. Genetic Cargo

Previous studies have shown that genetic materials were present in MVs isolated from Gram-positive bacteria [54,55]. With the continuous development of vesicle studies, it has been assessed that DNA of either chromosomal, plasmid, or phage origin and RNA (including mRNA, rRNA, sRNA, and tRNA) are carried through MVs [56,57]. These vesicles protect genetic material cargo from the degradation process through several molecules able to counteract extracellular nucleases [58,59]. A recent study on *Streptomyces coelicolor* showed that the DNA content carried by its MVs represents the entire chromosome of the bacterium [60]. Numerous studies have reported the association of DNA with bacterial MVs produced by Firmicutes. In *Ruminococcus* spp. strain YE71, the DNA in MVs was associated with short chromosomal fragments resistant to digestion, indicating differences in restriction/modification patterns compared to those in chromosomal DNA [61].

The majority of RNA in bacterial EVs tends to be relatively short and resistant to RNase treatment. For instance, miRNAs in *Streptococcus sanguinis* MVs are protected from degradation, ensuring safe transport to host cells [58]. In Group A *Streptococcus* EVs, Resch et al. observed the presence of many mRNA species, some of which were specifically enriched [24]. This suggests that EVs might induce the production of new proteins in recipient cells. These findings have significant implications for lateral gene transfer, potentially contributing to the spread of antibiotic resistance and virulence genes, as well as broader implications for bacterial evolution. Nevertheless, the specific mechanisms underlying the involvement of DNA and RNA remain unclear, warranting further investigation.

### 3.3. Virulence Factor Cargo

The exploration of cargo in pathogenic Gram-positive bacterial MVs has revealed the inclusion or association of virulence factors, suggesting a pivotal role in pathological conditions. Some of the MV protein cargo identified in Firmicutes encompasses enzymes engaged in peptidoglycan degradation, antibiotic degradation, and virulence factors (e.g., anthrolysin, anthrax toxin components, coagulases, hemolysins, and lipases). Wang et al. reported that macrophages activate their NLRP3 inflammasome when exposed to EV release containing pore-forming toxins and lipoproteins [62]. The polyketide toxin mycolactone, an important virulence factor of *M. ulcerans*, has been found in purified vesicles extracted from the abundant extracellular matrix that the bacteria use to infect humans, causing Buruli ulcer [63]. Similarly, MVs detected and purified from *Staphylococcus aureus* strains have been shown to contain biologically active toxins such as α- and γ-hemolysins, which disrupt eukaryotic cell membranes by pore formation, as well as exhibit superantigens (SEQ, SSaA1, and SSaA2) that can elicit proinflammatory mediators and cytotoxicity [20,23,64]. Analyses on *Listeria monocytogenes* MVs have revealed enrichment in proteins crucial for survival and virulence, including the hemolysin listeriolysin O [65]. MVs generated by Group A *Streptococcus* unveiled a set of 195 proteins in the EV proteome, including both distinctive proteins and those selectively enriched within the EVs [24].

## 4. Membrane Vesicle in Host–Pathogen Interactions

The role of vesicles in host–bacteria interactions is significantly determined by the cargo packaged during vesiculogenesis. The inclusion of toxins, siderophores, immune evasion proteins, adhesins, and antibiotic resistance proteins strongly suggests MVs’ involvement in virulence (Figure 2). Pathogenic bacteria can use MVs as a mechanism to deliver virulence factors to eukaryotic host cells. For instance, *S. aureus* MVs harbor superantigens, including T-cell-activating enterotoxin *SeQ*, lipase, immune evasion proteins (e.g., *protein A* and *SbI*), toxins such as PSMs, and bicomponent pore-forming toxins like alpha-toxin, *LukSF-PV*, and *LukAB*. MV-associated alpha-toxin from *S. aureus* significantly impacts certain diseases like atopic dermatitis more than its soluble forms [66]. Both forms induce keratinocyte death, but only MV-associated alpha-toxin provokes keratinocyte necrosis and eosinophilic infiltration specific to atopic dermatitis in mice [66]. Additionally, they may carry β-lactamase, which degrades β-lactam antibiotics, and staphopain A (a papain-like cysteine protease), contributing to extracellular matrix degradation and promoting tissue invasion [42,49,67]. Similarly, EVs from *L. monocytogenes* encapsulate the pore-forming toxin listeriolysin O, aiding the bacterium in escaping host vacuoles [25]. Notably, the cytosolic pore-forming toxin pneumolysin in *S. pneumoniae* lacks export signal sequences and is exclusively released into host cells through MV secretion, underscoring the pivotal role of MVs in *S. pneumoniae* virulence [47]. MVs enriched with extracellular-matrix-degrading enzymes from Group B streptococcus (GBS) play a role in disrupting physical barriers and causing host cell death. In mice, GBS-MV treatment led to collagen fragmentation, immune cell infiltration, and membrane integrity loss, resulting in preterm birth [28]. The heightened effectiveness of MV-associated toxins may be explained by their encapsulation within MVs. This allows toxins to be delivered at concentrated levels, avoiding dilution over a distance and providing protection from immune system clearance, including antibodies and protease activity. The differential impact of associated and soluble forms of toxins, whether MV-associated or soluble, is likely attributed to their distinct delivery mechanisms. The entry of membrane vesicles into eukaryotic host cells is a finely tuned process, involving various mechanisms tailored to the vesicle type recipient cell (Figure 3). The treatment of THP-1 cells or monocyte-derived macrophages with dynamin-dependent endocytosis prevents *S. aureus* MV internalization, hindering pore-forming toxin delivery [62]. Methyl-β-cyclodextrins abolished the internalization of EV-associated *protein A* from *S. aureus* inside human laryngeal carcinoma cells, potentially demonstrating that EVs fuse with cholesterol-rich domains of host cell membranes [64]. In *P. acnes*, clathrin-dependent endocytosis is the major route for MV internalization in human epidermal keratinocyte cells, with the specific receptor remaining unidentified [68]. Additional entry modes are probable, considering diverse routes described for OMVs, influenced by vesicle size and infected cell type [69].

## 5. Membrane Vesicles in Inter-Bacterial Interactions

Vesicle-mediated interactions among bacteria have been extensively explored in Gram-negative bacteria [70], particularly focusing on resistance gene transfer, bactericidal activities, and long-distance signaling within bacterial populations. Initially believed to be limited by thicker cell walls [1], MV-mediated processes in Gram-positive bacteria have gained attention, challenging prior assumptions. Well-known horizontal gene transfer (HGT) mechanisms, such as transformation, transduction, and conjugation, coexist with the proposed concept of “vesiduction”, which employs EVs in prokaryotes for the transfer of genes related to antibiotic resistance, virulence, and metabolic factors [55,71]. Recent studies, particularly in clinically significant species like *Staphylococcus aureus* [49] and *Enterococcus* spp. [72], have demonstrated the presence of MV-mediated mechanisms in Gram-positive bacteria. For instance, Lee et al. revealed the transfer of β-lactam antibiotic resistance from methicillin-resistant *S. aureus* ST541 to susceptible *E. coli* RC85 through MVs [73]. Contrary to previous beliefs, intracellular and extracellular DNA has been identified in Gram-positive bacteria, including *Streptococcus mutans* and *Clostridium perfringens* [21,56], challenging assumptions about MV-mediated genetic material transfer [74]. Despite uncertainties regarding determinant composition, MVs represent a potential strategy to mitigate the dissemination of multidrug-resistant (MDR) bacteria. *Streptomyces* has been found to frequently produce antimicrobial vesicles containing diverse compounds such as actinomycins, anthracyclines, candicidin, and actinorhodin, which exhibit diverse antibacterial and antifungal activity [75]. These antimicrobial-containing vesicles achieve direct delivery of their cargo to other microbes via membrane fusion, suggesting an alternative and broad delivery system for antimicrobial specialized metabolites with significant implications for interbacterial communication and new clinical strategies against antibiotic resistance. Additionally, vesicular inter-bacterial interactions extend to bactericidal activities, with Gram-positive vesicles commonly housing cell-wall-degrading enzymes [76]. MVs labeled with the lipophilic probe R18 from *B. subtilis* exhibit fusion capabilities with other *B. subtilis* cells, showcasing their broad interaction potential [77]. *Lactobacillus acidophilus* MVs seamlessly merge with membranes of *Lactobacillus delbrueckii* and *E. coli*, accompanied by the growth inhibition of target cells due to the presence of bacteriocins [78]. Further potential roles of MVs in inter-bacterial interactions could be represented by long-distance signaling within bacterial populations [79,80], as showed in research on Gram-negative bacteria. Specifically, studies on *Pseudomonas aeruginosa* have shown the use of its extracellular vesicles to transport PQS, a quorum sensing molecule. These vesicular PQS molecules can then directly interact with other bacteria through LPS [81]. The delivery of quorum sensing molecules embedded in MVs could specifically activate the recipient cells, leading to heterogenous gene activation within a bacterial population [81]. However, this aspect remains underexplored in the literature for Gram-positive bacteria; future studies have the potential to elucidate not only the transfer of resistance determinants but also the fundamental knowledge about the ways microbial communities interacts. 

## 6. Immune Regulation of MVs

The growing recognition of the role of MVs in host immune regulation is attributed to their content, which encompasses various immune-related molecules readily accepted by host cells [82]. These molecules, including lipoproteins and toxins, stimulate both innate and adaptive immunity through pattern recognition receptors (PRRs) [83]. MVs from Gram-positive bacteria, whether pathogenic or nonpathogenic, contribute to innate immunity involving key molecules like macrophages, dendritic cells, and Toll-like TLR2 [56,64,84]. The evolutionary costs of microbial MV immunogenicity must be balanced against the benefits to the microorganism from EV emission [85]. Additionally, MVs released by infected cells carry microbial molecules, potentially influencing the immune response indirectly [86]. Gram-positive bacterial EVs induce innate immune responses; for instance, MVs from *Clostridium perfringens* trigger the release of IL-6 through the TLR2 signaling pathway [56]. *Staphylococcus aureus* expresses co-stimulatory molecules via the TLR2 pathway, leading to the production of inflammatory proteins like tumor necrosis factor, IL-6, and IL-12, and releases EVs containing DNA, RNA, and peptidoglycans. These EVs are recognized by PRRs such as TLR7, TLR8, TLR9, and NOD2, ultimately undergoing autophagosomal degradation [64,87,88]. MVs from *Streptococcus suis* activate the nuclear factor kappa B signaling pathway, inducing the secretion of pro-inflammatory cytokines [84]. *Filifactor alocis* EVs elevate the secretion of various molecules in Thp-1 cells and oral keratinocyte cell lines [89]. Nonpathogenic bacteria, like *Lactobacillus sakei* subsp. *sakei* NBRC15893, show increased immunoglobulin A production through TLR2 signaling [90]. The cell wall components of Gram-positive bacteria, rich in TLR2 ligands, suggest that MVs may collaboratively exert innate immunomodulatory effects via TLR2 [91,92,93,94]. Lipoproteins within MVs play a role in immunomodulatory responses across various Gram-positive bacteria [90,95]. The TLR2-dependent pathway is crucial for the proinflammatory effect of EVs, with other pathways likely contributing. In the case of Gram-negative bacteria, various pathways leading to proinflammatory cytokine production, including TLR-4 and TLR-8 activation, the recognition of OMV-associated LPS and RNA, and NOD1 interaction with peptidoglycan, have been described [69,96]. *Pseudomonas aeruginosa* OMVs carrying sRNA52320 modulate the immune response by downregulating genes in the LPS-stimulated MAPK signaling pathway [97], suggesting potential interkingdom communication facilitated by RNA molecules in bacterial vesicles. It is probable that some of these mechanisms are shared and additional unique activation pathways exist for Gram-positive EVs. Additionally, anti-inflammatory potential has been predominantly observed in nonpathogenic species like *Lactobacillus paracasei*, demonstrating an in vivo impact against colitis [98]. EVs from other *Lactobacilli* spp. also dampen proinflammatory responses [99]. Surprisingly, EVs from the pathogenic bacterium *M. tuberculosis* exhibit anti-inflammatory activity by inhibiting CD4 T cell activation during macrophage infection [100]. Furthermore, MVs play a crucial role in adaptive immunity, with MVs from *Streptococcus pneumoniae* inducing specific antibodies [44], and *S. aureus* stimulating Th1, Th17, and Th2 cells, along with IgG antibody responses [87], suggesting the effective vaccination efficacy of MVs in Gram-positive bacteria.

## 7. Clinical Applications

Exploring bacterial MVs highlights their potential positive contributions to disease diagnosis and treatment. Current clinical applications of the utilization of MVs from Gram-positive bacteria have emerged as a promising therapy in several fields. Firstly, the immunity and stability of MVs highlight their potential as vaccine candidates. This is corroborated by evidence for major clinical Gram-positive pathogens. Choi et al. proposed how active immunization through *S. aureus* MVs induces an adaptive immunity of antibody and T cell responses. In particular, data showed that *S. aureus* MVs effectively protect against lethality and pneumonia induced by *S. aureus* infection, mainly via IFN-γ-producing Th1 cellular response rather than B-cell-mediated antibody response in mice [87]. Findings on *Streptococcus pneumoniae* revealed that extracellular MVs derived from *S. pneumoniae* BAA-255 effectively protected mice without causing significant side effects. Furthermore, the study identified specific immunogenic proteins present in these MVs. The results indicated that MVs, being more immunogenic than an equivalent bacterial cell extract, hold significant promise as potential vaccine antigens [101]. Moreover, Prados-Rosales et al. showed how MVs are highly immunogenic without adjuvants and elicit immune responses comparable to those achieved with BCG in protection against *Mycobacterium tuberculosis* [102]. To date, only one vaccine has been developed using OMVs for *Neisseria meningitidis* serogroup B: the MeNZB OMV vaccine. This vaccine was designed to provide broader protection against multiple strains of meningococcal B, reflecting advancements in vaccine technology to enhance its effectiveness [103]. However, further research should be carried out to pave the way for proper newly updated immunization therapies in Gram-positive bacteria.

Another clinically impactful theme is cancer therapy and the anti-tumor effects of bacterial MVs. Currently, the emphasis in cancer treatment development lies on targeted drugs; however, alternative approaches to identify and stimulate tumor immunity are also under investigation. Considering the immunomodulatory properties of MVs, they can be engineered to express cancer-specific epitopes or carry small noncoding RNAs, as previously explored for Gram-negative bacteria. Hence, OMVs have demonstrated the ability to induce a sustained antitumor immune response, effectively inhibiting tumor growth in various models [104]. For example, it has been shown that basic fibroblast growth factor (BFGF)-OMVs, used as vaccines, can successfully induce the body to produce persistent anti-BFGF autoantibodies to inhibit tumor growth and metastasis [105]. For Gram-positive bacteria, most of the evidence comes from several studies on probiotic strains belonging to the *Lactobacillus* genus, illustrating the existing approaches to regulate human gut microbial ecology, aiming at alternative methods to reduce the damage and improve the effectiveness of cancer therapy [106]. Notably, *Lactobacillus rhamnosus* GG, a commonly used probiotic supplement, generates MVs with cytotoxic effects on hepatic cancer cells. This is achieved, in part, through the downregulation of the *bcl-2* and *bax* genes in cancer cells [107]. Furthermore, MVs from *Lacticaseibacillus paracasei* PC-H1 can inhibit colorectal cancer cell growth both in vivo and in vitro, inducing apoptosis through the PDK1/AKT/Bcl-2 signaling pathway [108]. Gram-positive bacterial MVs may also play a role in various pathologies, including dermatological and viral diseases. *Streptococcus epidermidis* MVs show potential therapeutic value for psoriasis [109], while *Lactobacillus druckerii* MVs may have applications for hyperplastic scars [110]. In conclusion, Palomino et al. demonstrated that MVs from vaginally isolated Lactobacillus inhibited the adhesion and entry of the HIV-1 virus into human T cells [111] (Figure 4).

## 8. Future Directions

Engineered Bacterial Vesicles (BEVs) have emerged as versatile tools with promising applications across various domains, offering a glimpse into the future of medical and biotechnological advancements [112]. One of the most innovative applications lies in vaccine development, where BEVs could revolutionize the current approach to infectious diseases, providing a platform for the rapid development of effective vaccines [103,113,114,115]. Indeed, MVs can be engineered to carry specific antigens from pathogens, mimicking their natural presentation and potentially eliciting a more robust and targeted immune response [116].

In addition, BEVs hold great potential for drug delivery since they can be loaded with therapeutic agents such as antibiotics, antivirals, or immunomodulators, offering a controlled and targeted release system. This approach maximizes the efficacy of treatments, enhancing the stability of encapsulated cargoes [117] and at the same time minimizing side effects by directly delivering drugs to the site of infection [118]. Moreover, loaded BEVs preserve their immunogenicity, thereby simultaneously treating and preventing the infection. This combined effect may be helpful to counteract the spread of MDR bacteria. Since BEVs can contain a variety of bacterial biomolecules such as DNA, RNA, lipids, proteins, and metabolites, there is increasing evidence that they have significant potential as diagnostic biomarkers enabling the detection of early signs of infection and specifically identify the bacterial etiological agent [119]. The applications of BEVs extend beyond infectious diseases, opening new avenues for treating a wide range of diseases such as cancer or osteoporosis. In this context, they can be engineered to carry specific cargo, such as therapeutic proteins or small interfering RNA (siRNA) in order to modulate cellular processes for therapeutic purposes [120,121].

Moreover, there has been growing interest in investigating the impact of BEVs on Neurodegenerative Diseases (NDs) [122,123]. In recent years, research has highlighted the potential of BEVs to play a substantial role in the treatment and therapeutics of NDs [124]. Recent studies have suggested that these vehicles exhibit immuno-modulatory and neuroprotective properties, offering potential benefits in addressing NDs. Additionally, it has been noted that BEVs can effectively cross the Blood-Brain Barrier (BBB) and target specific regions of the brain, making them an appealing drug delivery system for ND treatment [124]. Despite their immense potential, engineered BEVs come with inherent challenges and limitations. Long-term studies are necessary to assess their safety and toxicity, especially as they progress toward clinical applications [125]. The standardization of production processes, scalability, and storage stability are critical obstacles that must be addressed to facilitate widespread use [126]. Due to their isolation from growth medium, batch-to-batch differences may occur, challenging current isolation methods to remove non-encapsulated bacterial components and discriminate between empty or filled vesicles [117].

## 9. Conclusions

There is a pressing need for innovative approaches to combat bacterial infections caused by Gram-positive bacteria. As far as we know, this study provides the first comprehensive review on how Gram-positive bacteria naturally interact with the host through MVs. Ongoing advancements in bacterial MV extraction technology have revealed the ability of Gram-positive bacteria to produce MVs, intricately linked to their virulence, immune capabilities, and pathogenic factors. While MVs assist bacteria in resisting host defenses, eliciting immune responses, and transferring drug-resistance genes, the molecular mechanisms remain unclear. MVs resist external environmental stresses, like antibiotic killing, suggesting a therapeutic strategy against antibiotic resistance. In addition, the immune response triggered by MVs, along with their adaptability for engineering, highlights their potential for various clinical applications. Probiotic MVs from Gram-positive bacteria hold promise for oral administration or incorporation into pharmaceutical supplements due to their regulatory and anti-inflammatory benefits. The prospect of revolutionizing vaccine development, drug delivery, diagnostics, and therapeutics is substantial, making BEVs a groundbreaking frontier in medicine and biotechnology. Nevertheless, practical implementation on a large scale requires addressing current limitations, ensuring safety, and developing robust production and regulatory frameworks. As research progresses, engineered MVs and their role in host–pathogen interactions hold the promise of transforming the healthcare and disease management landscape.

## Figures and Tables

**Figure 1 ijms-25-02904-f001:**
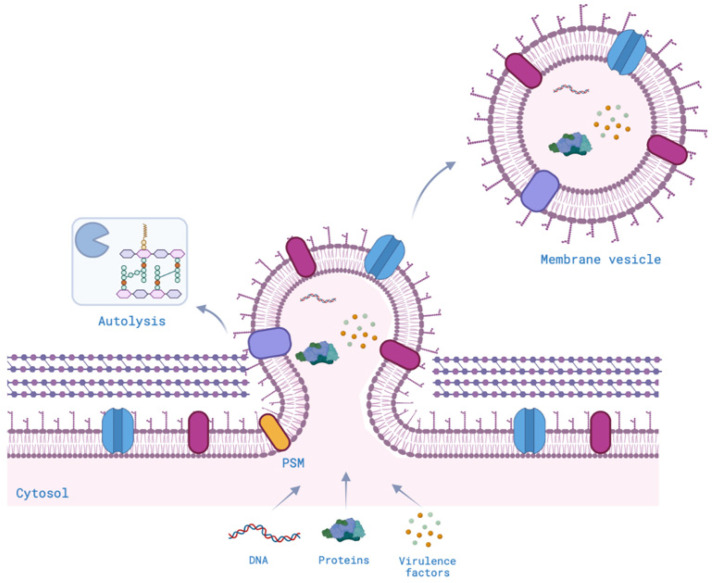
Vesiculogenesis of Gram-positive EVs (this figure was created by the authors using Biorender.com, app.biorender.com (accessed on 26 February 2024)).

**Figure 2 ijms-25-02904-f002:**
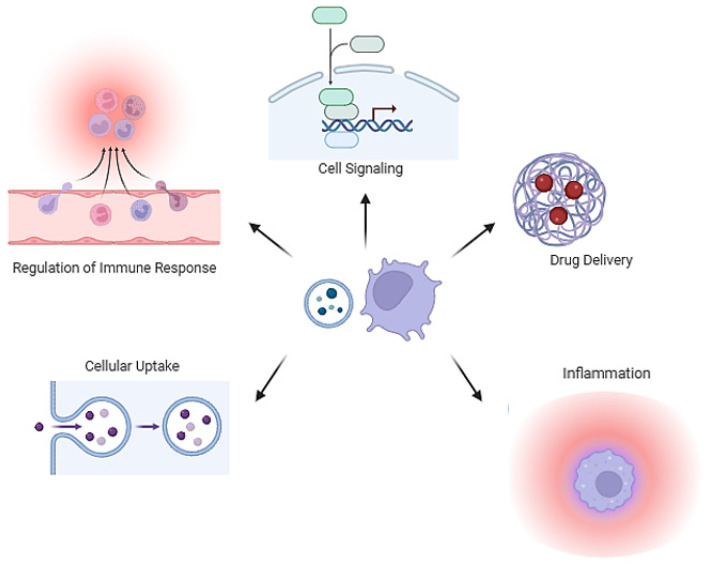
Example of possible interactions known between membrane vesicles and eukaryotic cells (this figure was created by the authors using Biorender.com, app.biorender.com (accessed on 26 February 2024)).

**Figure 3 ijms-25-02904-f003:**
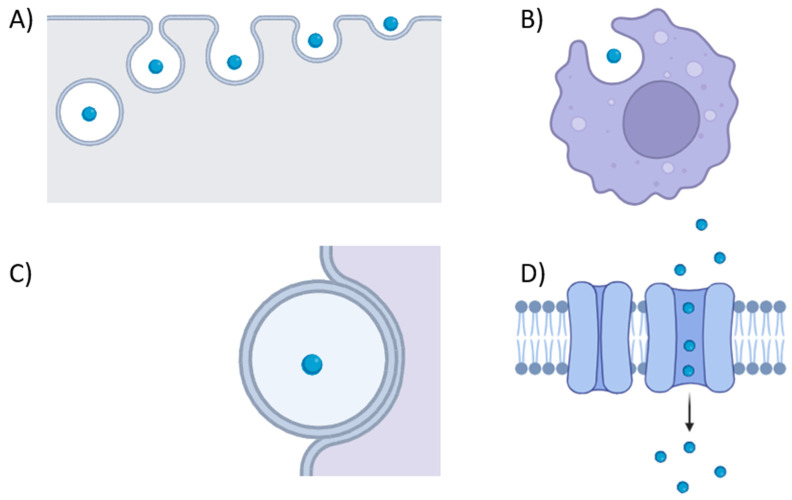
The different mechanisms of entry into host cells of membrane vesicles. EVs are represented as dark gray circles. Endocytosis (**A**), phagocytosis (**B**), fusion with membrane (**C**), and internalization mediated by receptor (**D**) (this figure was created by the authors using Biorender.com, app.biorender.com (accessed on 26 February 2024)).

**Figure 4 ijms-25-02904-f004:**
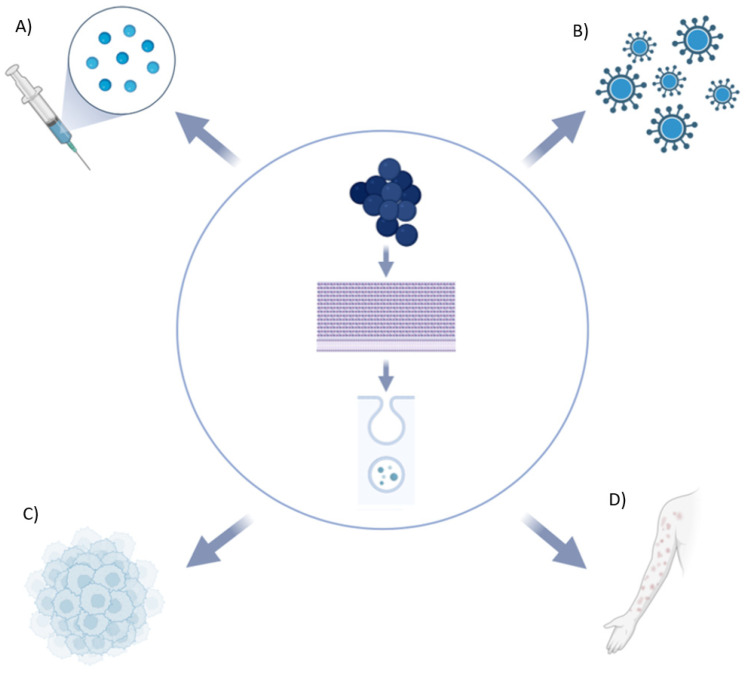
Gram-positive bacterial MVs and clinical applications: novel vaccination approaches (**A**), viral infections blocking (**B**), anti-tumor effects and cancer therapy (**C**), and dermatological diseases (**D**) (this figure was created by the authors using Biorender.com, app.biorender.com (accessed on 26 February 2024)).

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
