# Peer review of "Interactions of Gram-Positive Bacterial Membrane Vesicles and Hosts: Updates and Future Directions"

_ijms, 2024, doi:10.3390/ijms25052904_

Round 1

Reviewer 1 Report

Comments and Suggestions for Authors

This review provide an original and interesting focus on  membrane vesicles produced by Gram-positive bacteria, higliting their cargo and possible role in microbial relationships and interactions with other organisms.  The manuscript is well written and well structured. Below some suggestions, referring to specific lines (L), are provided to improve manuscript clarity and quality.

L41-43“Observations indicate the presence of MVs in both pathogenic and nonpathogenic Gram positive bacteria across diverse growth conditions and environments, suggesting the universal and widespread nature of MV secretion [16], [17], [18].” For a more complete description regarding the production of extracellular vesicles by non-pathogenic Gram-positive bacteria I would suggest including other examples, such as streptomycetes that are non-motile Gram-positive bacteria whose EVs may have a role in the relationships and interactions with plants, lower and higher animals, fungi and prokaryotes.

L46, L149, L232, L245 Please, use italics when appropriate (i.e. ok for genus and specie, no for spp., cells).

L54 Reference [22] does not focus on MV biogenesis. I would suggest indicating more appropriate references to be quoted with respect to MV biogenesis like references [1], [25] and [26] that seem more pertinent to me.

L72 In this line I would suggest including also [33] that is mentioned below in line 75.

L128-138 Some studiess (such as doi 10.1128/AEM.01881-21) focus on the fact that MV proteome can comprise not only proteins that reside inside the MV lumen and membrane-associated proteins but also proteins that are associated with MV surface. I would suggest mentioning this aspect for a more comprehensive description.

L139 “residing in either the lumen or the MV membrane”. Please, clarify how the genetic material resides in the MV membrane.

L144-147 It could be of some interest mentioning the presence of the whole genome in Streptomycete EV population (DOI: 10.1038/s41598-022-21002-z).

L206 Perhaps there is an extra “and” in the sentence.

L235 Please, rephrase. The sentence seems to affirm that “chromosomal” DNA has been identified in Streptococcus mutans and Clostridium perfringens.

L238-239 Please, improve clarity.

L255 It could be of some interest mentioning the presence of antibiotics in Streptomycete EVs (doi 10.1128/AEM.01881-21; doi 10.1128/jb.00325-23; doi 10.1016/j.chembiol.2017.08.008)

L265 Instead of the review [1] I would suggest citing the scientific work that specifically describes the study.

Figure 3 Legend should be improved. It should be indicated how the EVs are represented. In addition, panel C results quite confusing to me.

Figure 4 Image and legend should be improved. The central part of the figure “Gram-positive bacterial MVs …” should be described either in the legend and in the image.

Author Response

This review provide an original and interesting focus on  membrane vesicles produced by Gram-positive bacteria, highlighting their cargo and possible role in microbial relationships and interactions with other organisms.  The manuscript is well written and well structured. Below some suggestions, referring to specific lines (L), are provided to improve manuscript clarity and quality.

We want to thank you for your objective review, it was accurate and really helpful to us. Answers in bold. Here is how we addressed the typos:

L41-43 “Observations indicate the presence of MVs in both pathogenic and nonpathogenic Gram positive bacteria across diverse growth conditions and environments, suggesting the universal and widespread nature of MV secretion [16], [17], [18].” For a more complete description regarding the production of extracellular vesicles by non-pathogenic Gram-positive bacteria I would suggest including other examples, such as streptomycetes that are non-motile Gram-positive bacteria whose EVs may have a role in the relationships and interactions with plants, lower and higher animals, fungi and prokaryotes.

Thank you for your suggestion. In the specific lines, we added a citation (doi: 10.1128/AEM.01881-21) on Streptomycete EVs to implement the example of nonpathogenic bacteria.

L46, L149, L232, L245 Please, use italics when appropriate (i.e. ok for genus and specie, no for spp., cells).

Thank you for your feedback. We have made the necessary changes to utilize italics where appropriate.

L54 Reference [22] does not focus on MV biogenesis. I would suggest indicating more appropriate references to be quoted with respect to MV biogenesis like references [1], [25] and [26] that seem more pertinent to me.

Thank you for pointing this out. We have reviewed the references and have made the necessary updates to reflect more pertinent sources on MV biogenesis.

L72 In this line I would suggest including also [33] that is mentioned below in line 75.

Thank you for your suggestion. We have included reference [33] in line 72.

L128-138 Some studiess (such as doi 10.1128/AEM.01881-21) focus on the fact that MV proteome can comprise not only proteins that reside inside the MV lumen and membrane-associated proteins but also proteins that are associated with MV surface. I would suggest mentioning this aspect for a more comprehensive description.

Thank you for the suggestion. We have updated the paragraph (new lines 131-132) to include your suggestion.

L139 “residing in either the lumen or the MV membrane”. Please, clarify how the genetic material resides in the MV membrane.

Thank you for bringing this to our attention. We have revised the sentence to avoid misunderstandings.

L144-147 It could be of some interest mentioning the presence of the whole genome in Streptomycete EV population (DOI: 10.1038/s41598-022-21002-z).

Thank you for pointing out this relevant study. We have updated the paragraph (new lines 146-148) to include your suggestion.

L206 Perhaps there is an extra “and” in the sentence.

Thank you for the suggestion. We have corrected to improve clarity.

L235 Please, rephrase. The sentence seems to affirm that “chromosomal” DNA has been identified in Streptococcus mutans and Clostridium perfringens.

Thank you for pointing this out. We have rephrased the sentence.

L238-239 Please, improve clarity.

Thank you for the suggestion. We have corrected to improve clarity.

L255 It could be of some interest mentioning the presence of antibiotics in Streptomycete EVs (doi 10.1128/AEM.01881-21; doi 10.1128/jb.00325-23; doi 10.1016/j.chembiol.2017.08.008)

Thank you for pointing out these relevant studies. We have updated the paragraph (new lines 243-250) to include your suggestion.

L265 Instead of the review [1] I would suggest citing the scientific work that specifically describes the study.

Thank you for the suggestion.

Figure 3 Legend should be improved. It should be indicated how the EVs are represented. In addition, panel C results quite confusing to me.

Thank you for the suggestion. We have corrected to improve clarity.

Figure 4 Image and legend should be improved. The central part of the figure “Gram-positive bacterial MVs …” should be described either in the legend and in the image.

Thank you for the suggestion. We have corrected to improve clarity.

Reviewer 2 Report

Comments and Suggestions for Authors

The authors have provided a wide ranging review of Gram positive membrane vesicles. While there is little detail on or critical evaluation of the various papers cited, the review is more of a broad overview of the subject which is an appropriate approach for this review. The authors mention important implications of some of the findings, providing depth to the the review. I find the organization to be logical and easy to follow. There were a small number of typos:

Line 46: missing italics

Line 206: word "and" is superfluous? sentence doesn't make sense.

Line 208: missing italics

line 321: word "research" is missing?

Line 398: extra word "to"?

Author Response

The authors have provided a wide ranging review of Gram positive membrane vesicles. While there is little detail on or critical evaluation of the various papers cited, the review is more of a broad overview of the subject which is an appropriate approach for this review. The authors mention important implications of some of the findings, providing depth to the the review. I find the organization to be logical and easy to follow. There were a small number of typos:

Thank you for your feedback on our review. As you correctly guessed, we aimed to offer a broad overview of the subject, which we believe is appropriate for this review. We appreciate that you found the organization logical and easy to follow, and we are glad you found the implications of some of the findings to be important. Here is how we addressed the typos:

Line 46: missing italics

Thank you for your feedback. We have corrected the missing italics.

Line 206: word "and" is superfluous? sentence doesn't make sense.

Thank you for the suggestion. We have corrected to improve clarity.

Line 208: missing italics

Thank you for your feedback. We have corrected the missing italics.

line 321: word "research" is missing?

Thank you for bringing this to our attention. We have added the missing word.

Line 398: extra word "to"?

Thank you for bringing this to our attention. We have corrected to improve clarity.

Reviewer 3 Report

Comments and Suggestions for Authors

The review provides a thorough and well-articulated examination of the current knowledge surrounding MVs in Gram-positive bacteria, encompassing their content characterization, functions, and interactions with both host and bacterial cells. The authors offer detailed insights into the role of Membrane Vesicles (MVs) in Gram-positive bacteria, underscoring their significance in bacterial physiology and disease progression. While the manuscript is commendable, I have a few minor suggestions for improvement that the authors may wish to consider:

1.     Consider referencing reviews on Gram-negative bacteria in the introduction section when discussing research in this area.

2.     Figure 4 requires enhancement, particularly in terms of resolution.

3.     It may be beneficial to provide more descriptive figure legends for clarity.

Overall, I believe this review will capture the interest of a broad spectrum of scientists, and the authors deserve recognition for their commendable efforts in compiling this work.

Author Response

The review provides a thorough and well-articulated examination of the current knowledge surrounding MVs in Gram-positive bacteria, encompassing their content characterization, functions, and interactions with both host and bacterial cells. The authors offer detailed insights into the role of Membrane Vesicles (MVs) in Gram-positive bacteria, underscoring their significance in bacterial physiology and disease progression. While the manuscript is commendable, I have a few minor suggestions for improvement that the authors may wish to consider:

We want to thank you for your objective review, it was accurate and really helpful to us. Here is how we addressed the typos:

  1. Consider referencing reviews on Gram-negative bacteria in the introduction section when discussing research in this area.

Thank you for your suggestion. We have added another review on Gram-negative bacteria (doi: 10.1146/annurev-micro-052821-031444. in the section.

  1. Figure 4 requires enhancement, particularly in terms of resolution.

Thank you for your feedback. We have implemented Figure 4 to improve its clarity.

  1. It may be beneficial to provide more descriptive figure legends for clarity.

Thank you for your suggestion. We have revised the figure legends to improve clarity.

Overall, I believe this review will capture the interest of a broad spectrum of scientists, and the authors deserve recognition for their commendable efforts in compiling this work.